# FedEach: Federated Learning with Evaluator-Based Incentive Mechanism for Human Activity Recognition

**DOI:** 10.3390/s25123687

**Published:** 2025-06-12

**Authors:** Hyun Woo Lim, Sean Yonathan Tanjung, Ignatius Iwan, Bernardo Nugroho Yahya, Seok-Lyong Lee

**Affiliations:** Department of Industrial and Management Engineering, Hankuk University of Foreign Studies, Yongin 17035, Republic of Korea; 018lim@hanmail.net (H.W.L.); sea.remus@gmail.com (S.Y.T.); mania087@gmail.com (I.I.)

**Keywords:** incentive mechanism, federated learning, client selection

## Abstract

Federated learning (FL) is a decentralized approach that aims to establish a global model by aggregating updates from diverse clients without sharing their local data. However, the approach becomes complicated when Byzantine clients join with arbitrary manipulation, referred to as malicious clients. Classical techniques, such as Federated Averaging (FedAvg), are insufficient to incentivize reliable clients and discourage malicious clients. Other existing Byzantine FL schemes to address malicious clients are either incentive-reliable clients or need-to-provide server-labeled data as the public validation dataset, which increase time complexity. This study introduces a federated learning framework with an evaluator-based incentive mechanism (FedEach) that offers robustness with no dependency on server-labeled data. In this framework, we introduce evaluators and participants. Unlike the existing approaches, the server selects the evaluators and participants among the clients using model-based performance evaluation criteria such as test score and reputation. Afterward, the evaluators assess and evaluate whether a participant is reliable or malicious. Subsequently, the server exclusively aggregates models from these identified reliable participants and the evaluators for global model updates. After this aggregation, the server calculates each client’s contribution, prioritizing each client’s contribution to ensure the fair recognition of high-quality updates and penalizing malicious clients based on their contributions. Empirical evidence obtained from the performance in human activity recognition (HAR) datasets highlights FedEach’s effectiveness, especially in environments with a high presence of malicious clients. In addition, FedEach maintains computational efficiency so that it is reliable for efficient FL applications such as sensor-based HAR with wearable devices and mobile sensing.

## 1. Introduction

Federated learning (FL) has emerged as a machine learning paradigm by decentralizing the training process to a set of clients (e.g., mobile phones, IoT devices) without sharing raw or local data to accommodate the regulations (e.g., General Data Protection Regulation (GDPR)) [1] to safeguard privacy-concerned data. The FL process starts with the server distributing the current global model to clients. The clients employ their private training data to perform several steps and upload the update into the server. Subsequently, the server aggregates all the models uploaded by clients and broadcasts back the new global model to the clients to enable the next round of training. As initially proposed by Google in 2017, FL has been tested for distributed model training [2,3]. Despite its advantages over centralized methods, one of the biggest challenges is motivating clients to participate in FL by using their local data for model training. Without proper approaches, clients may be reluctant to contribute to the training process, which could cause slower convergence, lower accuracy, or even the failure of the FL system [4,5]. To keep the robustness of FL, some works proposed multiple aggregation rules such as Krum [6], Bulyan [7], trimmed mean [8], and median [8].

In real-world scenarios, there is a strong likelihood that disruptive clients will participate. This is a security threat known as Byzantine failures [9], where the clients do not rigorously follow the protocol and report arbitrary parameters to the server. Such disruptions can occur due to either faulty communication [10] or adversarial attacks, where malicious clients use corrupted data and report misleading vector updates and upload them into the server [11,12].

A common approach for addressing this issue is the incentive mechanism for client selections. It aims to filter out these malicious clients [13,14,15,16] by evaluating the contributions of the clients and applying penalties or incentives. By assessing contributions, the FL framework can both deter dishonest clients and encourage their long-term engagement [17].

In client selection with an incentive mechanism, three key aspects must be considered. First, it is essential to look for the clients’ contributions to the FL procedure. Clients’ contributions can be measured based on their impact on the global model’s accuracy, the consistency of their updates across multiple training rounds, and their adherence to the predefined training protocol. Properly evaluating contributions helps ensure that high-quality updates are aggregated, leading to a more robust and reliable FL system. However, many existing studies do not fully explore effective ways to assess and reward meaningful contributions, often emphasizing the removal of malicious clients instead [14]. Second, it is important to maintain performance in the presence of a large number of malicious clients. Many studies only deal with situations in which a small number of malicious clients attack [17,18], which may not reflect real-world conditions. Lastly, the performance of clients’ selection should have reasonable computational time. Many studies impose high computational cost by using data valuation methods that require evaluating all possible reliable clients [19,20].

Blockchain is considered an impressive technology when applied for an incentive mechanism in FL [13,14,19]. When combined with FL, blockchain enhances robustness and provides protection against counterfeiting. By recording the client activity for each round, smart contracts can automatically calculate and distribute incentives. However, the existing blockchain-based approaches have not explicitly addressed an FL environment with a high percentage of malicious clients (e.g., over 50% of the malicious clients exist). An existing work offers protection against up to 50% of the malicious clients [15] while others do not explicitly address this problem [21].

Common Byzantine FL schemes are categorized into Distance Statistical Aggregation (DSA) and Contribution Statistical Aggregation (CSA) [22]. DSA assumes that poisoning local models has a significant distance from benign local models and removes statistical outliers before aggregation [6,7]. Meanwhile, CSA prioritizes local models that contribute the most to the global model’s performance. Both approaches aim to identify and exclude the poisoned local model updates to ensure a high-quality global model. While these techniques primarily address attacks in FL, this study focuses on client selection. Combining CSA and DSA is expected to produce a robust blockchain-based client selection mechanism.

This study aims to propose a robust and computationally efficient incentive mechanism by utilizing blockchain techniques when it interacts with multiple clients. The proposed framework contains three modules: a training module, initialization module, and client selection module. At first, clients submit their model to the server using hash chain mechanisms (i.e., <key, values> pairs) and the initialization module attempts to select evaluators and participants. Afterward, the client selection module is executed by giving evaluators a role to evaluate the non-selected evaluators (i.e., participants) who could join the model aggregation by looking at the CSA along with extracting <key, values> pairs from the server. The contributions of this study are as follows:
We propose a novel evaluator-based client selection mechanism for FL. This design enhances robustness against malicious updates and improves computational efficiency by avoiding a centralized validation dataset.We present a method leveraging both DSA and CSA to address three important aspects of client selection:
o(1) Maintaining performance when the number of malicious clients exceeds 50% of the total;o(2) Prioritizing the client’s contribution by measuring the contribution of each client to the global serve;o(3) Minimizing computational costs up to twelve times compared to previous works with similar performance.We propose a client categorization mechanism that distinguishes between evaluators, who are designated to assess other clients within both the initial module and the client selection module, and reliable participants, who are identified as trustworthy based on their historical performance.

To evaluate the effectiveness of the proposed method, the performance of the proposed method is measured in a practical setting. One of the significant applications of federated learning (FL) is human activity recognition (HAR) where privacy is a major concern [23,24,25,26]. Previous studies on FL for HAR have primarily focused on FL architectures [23,27], personalization [24], limited labeled data [25], and constraint devices [26]. However, research on client selection in HAR remains limited. Client selection is crucial in HAR not only to mitigate attack or poisoning threats but also to enhance recognition accuracy. In addition, efficient client selection is important for HAR applications, where real-time processing and low computational cost are essential for practical solutions in resource-constrained environments such as wearable devices and mobile sensing platforms. The effectiveness of the proposed method is assessed across multiple critical aspects to ensure privacy preservation, guarantee reliable client selection, and enhance the feasibility of deploying client selection in FL-based HAR systems.

This study is organized as follows. Section 2 describes FL and related research on incentive mechanisms. The proposed method and its steps are proposed in Section 3. Section 4 is related to experiments. It shows the experiment setting and experiment results. In Section 5, we discuss the summary of this paper and the possible future work.

## 2. Related Work

This section delves into works that are relevant to this study, including FL and incentive mechanisms. First, we elaborate on the existing mechanism of FL, and we subsequently highlight the importance of incorporating incentive mechanisms within FL.

### 2.1. Federated Learning

Federated learning (FL) is a machine learning method that allows multiple devices to collaboratively train a comprehensive model [4]. Both server and clients can have a tuned machine learning model in the end without directly exchanging raw data, thus addressing the “isolated data” problem without risking data exposure. Instead of sharing data from each device, FL updates the overall model by merging the updates from every device. This decentralized approach to train a model offers benefits compared to traditional centralized machine learning. The overview of FL is shown at Figure 1. A study highlights the importance of compensation for clients to contribute [28]; otherwise, they might be disinclined to participate without compensation. Yet another study shows that even when compensated, clients could potentially submit inferior models [29].

There have been multiple existing works for incentive mechanisms to handle this problem [19]. According to an existing survey, the incentive mechanism is broken down into two categories by its design: a design driven by the client’s data contribution and a design driven by the client’s reputation. However, the work also shows that both designs recently incorporate a blockchain due to privacy issues when exchanging information such as data quality measures, data quantity measures, and reputation. This further highlights the importance of a blockchain-based incentive mechanism in FL.

### 2.2. Client Selection in FL

There are various approaches on client selection in FL [30,31]. In summary, there could be two categories: resource allocation and incentive/contribution mechanisms. The resource allocation mechanism attempts to look at client resources prior to the FL procedure to maintain the long-run process in FL. Meanwhile, the incentive/contribution mechanism considers the server resources to provide an incentive to clients and measure the contributions of the clients to improve the model performance.

#### 2.2.1. Resource Allocation Mechanism

One of the most common approaches in client selection is to select the clients based on the resources’ allocation. There was a work on selecting clients based on mobile edge computing resources prior to the FL procedure [32], called FedCS. It was proposed to aggregate updates by selecting clients with less resource constraints in one round. Another work proposed to group clients according to the given data classes and select one set in each communication round [33]. This mechanism averaged only similar clients and passed through different groups. In the domain of sensor-based HAR, many clients may have similar limited resources. Hence, it is unsuitable to perform this mechanism when the environment is similar.

#### 2.2.2. Incentive/Contribution Mechanism

The incentive mechanism aims to select clients with high-quality data based on a contract theory to join the FL procedure. Blockchain technology is often considered for incentivizing FL due to its inherent features that address transparency and trust issues [9]. Several studies have turned to cryptographic methods, including TIFF [34], BlockFlow [17], FIFL [21,28], and RRAFL [14].

The Shapley value is a well-known concept from cooperative game theory used to fairly evaluate each client’s contribution by measuring its marginal impact on the global model performance. In federated learning, several studies have adopted the Shapley value to enable fair and robust client selection.

ShapleyFL [35] is one example that computes a surrogate Shapley value (SSV) to adjust client weights and adopts importance sampling to preferentially select high-contribution clients that improve the robustness. However, this approach still relies on repeated model evaluations using a central validation dataset, resulting in significant computational overhead and limited scalability as the number of clients increases.

Block-RACS [18] extends this idea by combining a heuristic Shapley-based contribution index with a blockchain-based reputation and incentive mechanism. Clients bid for participation and are rewarded based on their estimated contribution. Although it achieves approximately 34% faster computation than an exact Shapley value, it still involves non-trivial computational overhead due to off-chain model evaluation and smart contract interactions. Additionally, it relies on benchmark datasets for contribution assessment, which may not always be practical in decentralized environments.

These methods utilize approximations of the Shapley value to reduce the computational cost of exact contribution evaluation. However, they still incur substantial computational overhead, particularly due to repeated model performance assessments. Moreover, they typically require access to a centralized benchmark or validation dataset.

Tokenized Incentive for Federated Learning (TIFF) [34] is the cryptographic techniques employed in incentive FL. In the TIFF approach, clients upload their local models along with their individual local test accuracy to the main server. The server saves accuracy and uses it when determining which clients will participate in training. Those selected receive rewards that correlate with the global model’s performance. While TIFF boasts efficiency as one of its merits, it did not highlight the problem of contaminated FL environment. Furthermore, it operates under the presumption that clients provide accurate reports of their results. Misrepresentations or exaggerated utilities from clients could adversely impact the global model’s performance.

Blockflow [17] operates as an incentive mechanism for federated learning, leveraging the public Ethereum blockchain. Within this framework, clients upload their local models and then assess the models of their peers using their individual data sets. The system identifies and penalizes dishonest clients by cross-checking with evaluation of other clients. Each client’s actions are recorded on the blockchain, and they’re compensated for their contributions. While Blockflow stands out for its capability to identify malicious clients, its efficiency holds primarily when the malicious participants represent less than half of the entire clients. When these malicious clients exceed 50%, the anticipated optimal outcomes also become doubtful.

A Fair Incentive Mechanism for Federated Learning (FIFL) [28] introduced an equitable incentive mechanism for FL. FIFL has an attack detection module which utilizes the gradients’ similarity distance between the benchmark gradient and each client’s gradient. If this difference exceeds a predetermined threshold, that particular client is labeled as a malicious client. Outcomes from the detection module are recorded in a reputation module after every round. It encourages clients to maintain honesty throughout multiple rounds, not just a singular one. While FIFL proves efficient, its dependence on a benchmark gradient stands as a notable constraint. It also has not addressed the problem of a contaminated FL environment.

RRAFL [14] is an incentive-based FL mechanism that integrates reputation and reverse auction principles. Initially, the task requester (server) announces task details, prompting clients to place bids. Based on these bids and the reputations of the clients, the requester chooses which clients will be participating in the training. Within RRAFL, a client’s reputation is calculated by both the server and other clients. RRAFL meets rationality and honesty. However, the server needs pre-training steps to provide prior reputation parameters in the first round. Furthermore, if more than 50% of the clients have malicious intent, the system’s robustness can be diminished.

The earlier methods have limitations on the computational cost and robustness. When there are a vast number of clients, this requires a reasonable computational cost. In the meantime, early methods have reliable performance but high computational costs. In addition, early methods considered only less than 50% of the malicious clients while the real-world environment may exist as more than 50%. There should be a robust mechanism against most malicious clients (i.e., the existence of more than 50% of the malicious clients).

## 3. Proposed Methodology—FedEach

This section introduces the proposed framework, called FedEach. The framework contains three modules: a training module, initialization module, and client selection module. The training module aims to trigger the clients to carry out the local training. The initialization module is a module that runs only on the initial round. It aims to determine the initial evaluators and participants since there is no information about the clients’ reputations and contributions. Lastly, the client selection module aims to aggregate the global model, measure the metrics (i.e., reputation and contributions), and select the new evaluators and participants for the next rounds. Figure 2 is a diagram of the flow in the FedEach framework.

### 3.1. Training Module

In federated learning, the initial stages are varied. Generally, it starts with a trigger from the server to request local training for the clients. In the general scheme of FL, all clients are considered as equal. However, in FedEach, all clients participate in the training in the initial round, but only selected clients can participate thereafter. All the notations used for our framework can be seen in Table 1.

Initially, the server lists a set of n clients $C=\{c_{1},c_{2},..,c_{n}\}$ and clients’ local models $M=\{m_{c_{1}},m_{c_{2}},..,m_{c_{n}}\}$. The server (i.e., the proposer in Paxos, with a prepare request mechanism for n clients) requests clients to carry out local training and submit their local models into the server. At the initial round, there is no information about the client, so all the clients participate in the training without any conditions. Hence, all the clients undertake local training and submit their local model to the server.

This module also aims to request the clients to carry out the local training on the other rounds. In the other rounds, the clients also need to submit their local model into the server with hash values. The difference between the initial round and the other round is the provision of the metrics (e.g., reputation) that would be of worth for the evaluators and participant selections. After the initial round, according to the metrics, only clients selected as evaluators or participants undertake local training.

The server stores the client’s model with a hash value for securing the identification of clients when other clients test their local models. In other words, the use of the hash value could prevent identity leakage.

### 3.2. Initialization Module

To select the earliest evaluators and participants, an initialization module is needed in the initial round since there is no information about the clients’ reputations and contributions. It should be noted that the initialization module is only implemented in the initial round and not thereafter (refer to Algorithm 1).
**Algorithm 1:** Initialization Module**Input****:**   *k*, β**Output:**  
$R_{c_{i}},T,Z$
1:**for**$\;\mathrm{each}\;c_{i}$ ∈ C do2:  **for**$\;\mathrm{each}\;m_{c_{j}}$(*i* ≠ *j*)3:   Tests and report $\theta_{c_{i},c_{j}}$ to the server4:
  **end for**
5:**end for**6:Calculates Sci and appends Sci in Rci for each client7:Selects top k evaluators with the high SciT={t1,t2,...,tk}V=v1,v2,...,vn−k=C∖T8:Calculates  Pvi0 for each viPvi0 = Svi/∑u=1|V|Svu9:Selects β participants according to  Pvi0 among ***V***

$Z=\left\{z_{1},z_{2},...,z_{\beta }\right\}$

After the training module, each client $c_{i}$ loads the other clients’ models ($m_{c_{j}},\forall j\;⋀\;i\neq \;j$) with a hash value and tests the other clients’ models with their own local dataset. After testing, the client submits the test scores ($\theta_{c_{i},c_{j}}$) to the server. $\theta_{c_{i},c_{j}}$ is the test score of client $c_{i}$ tested by client $c_{j}$ with $c_{j}$’s local data. For $\theta_{c_{i},c_{j}}$, metrics like the F1-score or accuracy can be used for classification challenges.

The server collects every individual $\theta_{c_{i},c_{j}}$ and averages them to generate $S_{c_{i}}$. Using $\theta_{c_{i},c_{j}}$, the server computes the averaged score ($S_{c_{i}}$) according to Formula (3). *n* is the number of clients. $\Theta_{c_{i},C\;}$ is {$\theta_{c_{i},\;c_{1}},\theta_{c_{i},\;c_{2}},\theta_{c_{i},\;c_{3}},\;\ldots .\;,\;\theta_{c_{i},\;c_{n}}\}\setminus \theta_{c_{i},\;c_{i}\;}.$ $S_{c_{i}}$ is divided into two components. $U_{c_{i}}$ represents the average of $\Theta_{c_{i},C\;}$. $Q_{c_{i}}$ imposes a penalty based on the difference between the median of $\Theta_{c_{q},\;C\;}$ and $c_{i}$’s submitted test score ($\theta_{c_{q},c_{i}}$) for each client.(1)$$U_{c_{i}}=\frac{1}{n-1}\sum_{j=1}^{n}\theta_{c_{i},c_{j}},\;(i\;\neq j).$$
(2)$$Q_{c_{i}}=\frac{1}{n-1}\sum_{q=1}^{n}max(0,median(\Theta_{c_{q},C})-\theta_{c_{q},c_{i}}),(i\;\neq q).$$
(3)$$S_{c_{i}}=U_{c_{i}}+\max\left(0,{\;U}_{c_{i}}-Q_{c_{i}}\right).$$

From the set of clients, the top *k* clients based on their $S_{c_{i}}$ are designated as evaluators, $T=\{t_{1},t_{2},...,t_{k}\}\subset C$, while the others are candidates, $V=\left\{v_{1},v_{2},...,v_{n-k}\right\}=C-T$. $S_{c_{i}}$ is appended to $c_{i}$’s reputation ($R_{c_{i}}),$ which will be explained later in Section 3.3.2.

Let $Z=\left\{z_{1},z_{2},...,z_{\beta }\right\}\;\subset \;V$ be the participants with *β* as the count of participants. The participants are selected among the candidates based on the importance ratio ($P_{v_{i}}^{r}$). $P_{v_{i}}^{r}$ represents the probability that candidate *v_i_* will be selected as a participant in a particular round (*r*). Note that the participant selection would be based on roulette wheel selection. This means that the higher probability of the candidate refers to the higher chance of the candidates being selected as participants. The calculation method for $P_{v_{i}}^{r}$ changes depending on the specific round. In the initial round (round zero), the importance ratio of $v_{i}$ ($P_{v_{i}}^{0}$) is shown in Formula (4) below:(4)$${\;P}_{v_{i}}^{0}=\;S_{v_{i}}/\sum_{u=1}^{\left|V\right|}S_{v_{u}}.$$

Regarding the calculation in the other round, this will be explained in Section 3.3.2.

### 3.3. Client Selection Module

The client selection module is designed to select clients for aggregation, assess them, and record their contributions. This module aims to filter out malicious clients and fairly assess client contributions. This module contains two main steps: (1) evaluation and (2) contribution measurement and reputation. The evaluation step is to determine the reliable and malicious participants. Subsequently, the contribution measurement and reputation step is to assess the clients who join the aggregation (i.e., evaluators and reliable participants) and record their contributions’ score and reputations’ score.

#### 3.3.1. Evaluation

The first step of the client selection module is an evaluation. This step is conducted by evaluators (***T***) on participants (***Z***) to distinguish whether the participant is reliable or malicious. As a result of the evaluation, this can prevent malicious participants’ models from influencing the global model.

The evaluators are required to load the participants’ local models (with a hash value) and then evaluate these models using their own local data. After the evaluation, evaluators submit the evaluation result ($E_{z_{i},\;t_{j}}$) to the server. $E_{z_{i},\;t_{j}}$ is the evaluation result of participant $z_{i}$ by evaluator $t_{j}$ with $t_{j}$’ s local data. Various metrics like the F1-score and accuracy can be used for $E_{z_{i},\;t_{j}}$.(5)$$\mathrm{avg}E_{z_{i},\;t_{j}}=\frac{\sum_{j=1}^{\left|T\right|}E_{z_{i},\;t_{j}}}{k}.$$

To generalize the participants’ performance, the server computes the average of the evaluation result for each participant (avg$E_{z_{i},\;t_{j}}$). If avg$E_{z_{i},\;t_{j}}$ surpasses threshold(δ), participant $z_{i}$ is identified as a reliable participant, $z_{i}$ ∈ ***X***. Only those models belonging to reliable participants and evaluators can be used in the aggregation process. The local models of reliable participant $x_{i}$ and evaluator $t_{j}$ are denoted as $m_{x_{i}}$ and $m_{t_{j}}$ ($m_{x_{i}}$, $m_{t_{j}}\;\epsilon \;M$), respectively. Meanwhile, the global model is marked as ${m^{\prime }}_{g}.$ A new global model can be derived as Formula (6).(6)$${m^{\prime }}_{g}=\frac{1}{\alpha }\sum_{i=1}^{\left|X\right|}m_{x_{i}}+\frac{1}{k}\sum_{j=1}^{\left|T\right|}\;m_{t_{j}}.$$

The overall evaluation and aggregation can be seen in Algorithm 2. The client categorization mechanism can be seen in Figure 3.
**Algorithm 2:** Evaluation and Aggregation**Input:**  *δ***Output:**
  ${m^{\prime }}_{g}$
1:**for**$\;\mathrm{each}\;\mathrm{participant}\;z_{i}$ do2:
  $\mathrm{Train}\;\mathrm{local}\;\mathrm{model}\;m_{z_{i}}$
3:**end for**4:**for** $\mathrm{each}\;\mathrm{evaluator}\;t_{j}$ do5:
  **for**
$\;\mathrm{each}\;m_{z_{i}}$
6:
   $\mathrm{Evaluates}\;\mathrm{and}\;\mathrm{submits}\;E_{z_{i},\;t_{j}}$
7:
  **end for**
8:**end for**9:Calculates avg$E_{z_{i},\;t_{j}}$
10:**if** avg$E_{z_{i},\;t_{j}}$11:  $z_{i}$ → X
12:**else**13:
  Append 0 
in Rzi
14:**end if**15:Sever aggregates local models of T and Xm′g = 1α∑i=1|X|mxi + 1k∑j=1|T|mtj

#### 3.3.2. Contribution Measurement and Reputation

This step aims to assess clients’ contributions. In the contribution measurement, only reliable participants and evaluators can participate after the aggregation process. By DSA, let ${sim}_{c_{i},r}$ be the contribution degree of $c_{i}$ in a particular round (r). To calculate ${sim}_{c_{i},r},$ FedEach measures the similarity between $m_{c_{i}}$ and ${m^{\prime }}_{g}$. ${sim}_{c_{i},r}$ ranges between [−1, 1], where a value closer to 1 signifies a high similarity to the global model. It can be explained that the low ${sim}_{c_{i},r}$ implies the low contribution, whereas a high ${sim}_{c_{i},r}$ represents a high contribution towards the global model. Initially, both $m_{c_{i}}$ and ${m^{\prime }}_{g}$ are flattened into one-dimensional vectors and calculate the cosine similarity, ${sim}_{c_{i},r}$, between the two vectors following Formula (7) [16]:(7)$${{sim}_{c_{i},r}=m}_{c_{i}}\cdot \;{m^{\prime }}_{g}/\;||m_{c_{i}}\left||\;*\;||{m^{\prime }}_{g}|\right|.$$

The result of the contribution measurement (${sim}_{c_{i},r}$) is appended to each client’s reputation ($R_{c_{i}}$). $R_{c_{i}}$ is a list, which is stored in smart contract on the server, to record $c_{i}$’s contribution for each round. It is composed of $S_{c_{i}}$ and ${sim}_{c_{i},r}$. The structure of $R_{c_{i}}$ is as follows:(8)$$R_{c_{i}}=\;[S_{c_{i}},\;{sim}_{c_{i},0},\;{sim}_{c_{i},1},\;\ldots .,\;{sim}_{c_{i},r-1},\;{sim}_{c_{i},r}]$$

Note that the reliable participant is evaluated, and the contribution score is recorded as well. For the malicious participants, the value of avgEzi, tj is lower than δ (the given threshold), and zero is added to Rzi. For non-participants (V\Z), simci,r is set to null. The top *k* clients based on their simci,r are designated as the evaluators for the next round.(9)Pvir=∑q=1wRvi[|Rvi|−w+q]∑u=1|V|∑q=1wRvu[|Rvu|−w+q], (r≠0),w=min(r,w).

At the beginning of the next round (except round zero), the server selects new participants based on the importance ratio. After the initial round, we redefine the importance ratio of $v_{i}$ ($\;P_{v_{i}}^{r}\;$) as shown in Formula (9). For fairly selecting the participants and improving the selection’s efficiency, it is necessary to select the ${sim}_{c_{i},r}$ from the latest round. For example, when the number of rounds is 1000, the selection criteria could be based on the latest 10 rounds. However, participants who could not contribute consecutively on every round can be eliminated gradually. For example, participants which have ${sim}_{c_{i},r}$ as zero for five consecutive rounds will not be selected anymore for future rounds. As a result, malicious candidates can be gradually eliminated.

For the computation of $P_{v_{i}}^{r}$, it utilizes the moving average on which w refers to the window size. The server calculates the moving average of $R_{v_{i}}$ and uses the last elements of $R_{v_{i}}$ to compute $P_{v_{i}}^{r}$. Note that w will be equal to r when w is bigger than r. By CSA, candidates who maintain a strong $P_{v_{i}}^{r}$ stand a higher chance of selection but those with a low $P_{v_{i}}^{r}$ also obtain a few chances to participate in the training. This prevents overfitting and gives a chance to clients who have low resources.

The server selects new participants based on $P_{v_{i}}^{r}$ and distributes a new global model to new evaluators and participants. After distribution, repeat the training module and client selection module until all the processes are completed.

After every round is over, the incentive is calculated with a smart contract on the server (Task 10). Clients with a sum of contribution (${contri}_{c_{i}}$) exceeding the predefined criteria could receive utility (${util}_{c_{i}}$) as incentives. Note that the criteria is set before the training and criteria ∊ [0, max(${contri}_{c_{i}}$)]. The notation ${util}_{c_{i}}$ is denoted as Formula (11):(10)$${{contri}_{c_{i}}=S}_{c_{i}}+\sum_{h=0}^{\left|\Gamma \right|-1}{sim}_{c_{i},h}.$$(11)$${util}_{c_{i}}=\phi ({contri}_{c_{i}},\;criteria).$$

The overall contribution measurement and reputation mechanism can be seen in Algorithm 3.
**Algorithm 3:** Contribution Measurement and Reputation**Input****:**  *w*, *k*,β**Output:**  
$R_{x_{i}}$
$,\;R_{t_{j}}$
$,\;T^{\prime }$
**,**
$Z^{\prime }$
1:**for** 
$x_{i}$
*,*
$t_{j}$
 in X
**,**
T
2:  ${sim}_{x_{i},r}$ = $m_{x_{i}}$· ${m^{\prime }}_{g}$/ |$|m_{x_{i}}$|| * ||${m^{\prime }}_{g}||$
3:  ${sim}_{t_{j},r}$ = $m_{t_{j}}$· ${m^{\prime }}_{g}$/ |$|m_{t_{j}}$|| * ||${m^{\prime }}_{g}||$
4:
  $\mathrm{Append}\;{sim}_{x_{i},r}\;in\;R_{x_{i}}$
5:
  $\mathrm{Append}\;{sim}_{t_{j},r}\;in\;R_{t_{j}}$
6:**end for**7:$\mathrm{Select}\;\mathrm{new}\;\mathrm{top}\;k\;\mathrm{evaluators}\;\mathrm{with}\;\mathrm{the}\;\mathrm{high}{\;sim}_{x_{i},r}$ 
$\mathrm{or}\;{sim}_{t_{j},r}$
$T^{\prime }=\{{t^{\prime }}_{1},{t^{\prime }}_{2},...,{t^{\prime }}_{k}\}$
$V^{\prime }=\left\{{v^{\prime }}_{1},{v^{\prime }}_{2},...,{v^{\prime }}_{n-k}\right\}=C\setminus T^{\prime }$
8:$\mathrm{Calculates}\;P_{v_{i}}^{r}$ 
$\mathrm{for}\;\mathrm{each}\;{v^{\prime }}_{i}$
$P_{v_{i}}^{r}$ *=* 
$\frac{\sum_{q=1}^{w}R_{v_{i}}[\;|R_{v_{i}}|-w+q]}{\sum_{u=1}^{\left|V\right|}\sum_{q=1}^{w}R_{v_{u}}[\;|R_{v_{u}}|-w+q]}$
9:Selects new β 
$\mathrm{participants}\;\mathrm{according}\;\mathrm{to}\;P_{v_{i}}^{r}$
$\mathrm{among}\;V^{\prime }$
$Z^{\prime }=\left\{{z^{\prime }}_{1},{z^{\prime }}_{2},...,{z^{\prime }}_{\beta }\right\}$


## 4. Experiment

This section conveys the experiment of the proposed framework. To show the effectiveness of the proposed framework, three datasets of human activity recognition (HAR) are used. The experimental setting is explained to follow the flow in the proposed framework. At the end, the experiment results for fairness, robustness, efficiency, and the sensitivity experiment for threshold (δ) are explained.

### 4.1. Dataset Introduction

The dataset for the experiment include three datasets: UCI-HAR [36], USC-HAD [12], and WISDM [10]. All datasets are in the domain of sensor-based HAR.
UCI-HAR. The dataset comprises 30 subjects carrying waist-mounted smartphones with embedded inertial sensors. The dataset is labeled with six activity classes: walking, walking upstairs, walking downstairs, sitting, standing, and laying.USC-HAD. The dataset contains 12 activity classes including walking, running, and stair ascent and descent. The activities were performed by a group of 14 volunteers, and the data were collected using a MotionNode sensor that was placed on the front right hip of each volunteer.WISDM. The dataset consists of 18 activities, which are categorized into six classes such as non-hand-oriented activities and hand-oriented activities. Fourteen subjects perform activities like walking, jogging, and kicking a ball. The dataset was compiled using accelerometer and gyroscope data from a smartphone, which was kept in the right pocket of the volunteer’s pants, and a smartwatch, which was worn on the volunteer’s dominant hand.

### 4.2. Experimental Setting

For the dataset preprocessing, we performed windowing with 80% overlap for WISDM and 50% for UCI-HAR and USC-HAD. In UCI-HAR, the data are already windowed for 2.56 s with 128 readings according to the existing literature. For USC-HAD, we resampled the data into 100 Hz and windowed them in 5 s. Meanwhile, for WISDM, we resampled the data into 20 Hz and windowed them in 10 s.

We distributed the dataset equally among all clients and split it into 70–30, thereby being a training–evaluation dataset in which each subject represents a client. The evaluation metric is the F1-score for $\theta_{c_{i},c_{j}},\;E_{z_{i},\;t_{j}}$. The number of evaluators (k) is two and the number of participants (β) is 50% of the total number of clients.

For the experiment, we choose random clients as malicious clients and add Gaussian Noise to the training data on those clients. The proposed method is evaluated on three aspects: fairness, robustness, and efficiency. We use a simple CNN model with three layers. The experiment is repeated five times, and the averages of the performances are taken. Each experiment iteration records model performance during 50 rounds and sets three local epochs for each round.

The hardware environment is an 11th Generation Intel Core i7-1165G7 processor, supplemented by 16 GB of RAM and an NVIDIA GeForce MX450 graphics card. The codebase was created using Python version 3.9.7 and developed within Jupyter Notebook (7.3) and Visual Studio Code (2022) environments. The PyTorch library, version 1.11.0, was utilized for this study.

There are several methods for comparison. The baseline is Federated Averaging (FedAvg) [3], where every client participates in aggregation without any conditions. The second method is Shapley Value Selective Federated Learning FL (SVS) [19]. In SVS, the server calculates the Shapley value of clients. The clients whose Shapley value exceeds the threshold can participate in the aggregation process. The last method is Tokenized Incentive for Federated Learning (TIFF) [34]. In TIFF, each client trains local models and reports the model parameters and the accuracy of the local model. Subsequently, the server selects the top $\frac{k+\beta \;}{2}$ clients with high accuracy and selects $\frac{k+\beta \;}{2}$ clients randomly in every round. Only selected clients can join aggregation to build the global model.

### 4.3. Experimental Result

This section addresses three kinds of evaluations: fairness, robustness, and efficiency. Fairness aims to look for clients who provide high data quality. Robustness defines performance stability even if there are more than 50% of the malicious clients. Efficiency refers to the computational time of the proposed method. The sensitivity experiment for the threshold (δ) aims to analyze the effect of the value of the threshold against the performance of the framework.

#### 4.3.1. Fairness

To validate the fairness of FedEach, we checked ${util}_{c_{i}}$ of subject #1 according to the noise level.

In this experiment, ${util}_{c_{i}}$ is defined as follows (12) and the criteria is one.(12)$${util}_{c_{i}}=\max(\;{contri}_{c_{i}}-\mathrm{criteria},0).$$

As aforementioned, δ presents the threshold to categorize the participants into reliable and malicious clients. δ may be set in consideration of data characteristics, e.g., Homogeneity and robustness againts noise, etc. δ of UCI-HAR is 0.7 and the others are 0.5. This is because UCI-HAR is more homogenous than the others.

When δ equals zero, models from all participants are incorporated during aggregation. When δ is zero, if the noise level increases, ${util}_{c_{i}}$ diminishes. This decline can be attributed to clients with higher noise levels obtaining a lower ${sim}_{c_{i},r}\;$ value. A reduced ${sim}_{c_{i},r}$ value subsequently leads to a decrease in $P_{c_{i}}^{r}$, causing their ${util}_{c_{i}}$ to shrink. Thus, it effectively shows that FedEach evaluates the contributions of clients fairly.

When δ is non-zero, FedEach can filter out malicious clients. Only models which are qualified in evaluation are utilized for aggregation. If the evaluation is not passed, ${sim}_{c_{i},r}$ becomes 0. Not only does it reduce $P_{c_{i}}^{r}$, but it also ultimately reduces ${contri}_{c_{i}}$. According to the util function used in the experiment, incentives are not received if ${contri}_{c_{i}}$ is lower than the criteria. In Figure 4, if the noise level rises, ${util}_{c_{i}}$ decreases, and if it rises above a certain level, ${util}_{c_{i}}$ converges to zero. Therefore, FedEach can distribute incentives fairly according to data quality.

#### 4.3.2. Robustness

To check robustness, we observed accuracy in two situations where malicious clients are less than 50% and more than 50% of the total clients. The noise level of malicious clients is 0.5.

As shown in Table 2, while FedAvg shows lower performance, most methods remain effective when malicious clients are 20% of the total. However, when the proportion of malicious clients increases to 70%, the results change significantly. The accuracy of FedAvg and TIFF drops by at least 15%, highlighting their vulnerability in high-adversary settings. In contrast, FedEach and SVS demonstrate greater robustness even when malicious clients exceed half of total participants. Note that we report results for 20% and 70% of the malicious clients to illustrate the low- and high-adversarial scenarios, as intermediate cases showed unstable performance with high variability, which could obscure the overall trend. The underlined result showed the highest accuracy among the experiments on different methods.

#### 4.3.3. Efficiency

To check efficiency, we observed the running time of three client selection methods with USC-HAD.

For this experiment, the global round is 10 and the number of clients is 11. In earlier experimental findings, both FedEach and SVS exhibited robust performance despite the presence of several malicious clients. However, a significant disparity in their efficiency was observed. Table 3 shows that TIFF is the most efficient, but as we have seen in previous experiments, TIFF underperforms when there are multiple malicious clients. Both SVS and FedEach are robust, but SVS’s running time is approximately 12 times larger than that of FedEach. Therefore, only FedEach satisfies both efficiency and robustness.

#### 4.3.4. Sensitivity Experiment for Threshold (δ)

As aforementioned, δ should be set in this consideration. High δ causes it to be difficult for participants to pass the evaluation. Thus, participants with relatively considerable data and a model might not pass the evaluation. To explore the impact of δ, we observed accuracy depending on the noise level and δ with the USC-HAD dataset.

According to Figure 5, when the noise level is between 0.1 and 0.3, the accuracy decreases as δ increases. It shows that when the noise level is relatively low, the quantity of data plays a more significant role than selecting high-quality clients, as it is robust against low levels of noise.

When the noise level is greater than 0.5, the accuracy decreases if δ becomes higher than 0.5. Because it is hard to maintain performance with a high level of noise, removing malicious participants diligently with high δ may increase the performance. However, if δ is more than 0.5, the accuracy decreases again. This displays how excessive δ can rather deteriorate the performance.

### 4.4. Discussion

This study introduces FedEach, a client selection framework designed to enhance robustness in FL. Unlike existing approaches such as FedAvg and TIFF, FedEach incorporates a client selection mechanism that verifies client reliability through cross-comparison strategies and reputation tracking which reflect their past performance. This approach enhances robustness against malicious clients and operates effectively in scenarios where over 50% of the participating clients behave adversarially. Note that FedEach achieves this without the need for a centralized validation dataset, which is often impractical in privacy-preserving FL settings.

Compared to SVS, a state-of-the-art method for fair client contribution evaluation, FedEach provides more computational efficiency. While SVS relies on sampling-based marginal contribution estimation (inspired by the Shapley value), it suffers from high computational costs as the number of clients and training rounds increases. In contrast, FedEach achieves comparable fairness in client contribution evaluation through lightweight scoring techniques such as Distance Statistical Aggregation (DSA) and Contribution Statistical Aggregation (CSA) (Section 3.3.1 and Section 3.3.2), making it more suitable for real-world FL applications with limited resources.

Nevertheless, FedEach is not without limitations. The inclusion of additional processes, such as evaluator selection and scoring, introduces extra computational steps in each global round, which may increase overall complexity compared to simpler baselines like FedAvg. Furthermore, the threshold parameter (δ) used in determining client reliability must be manually configured, which may require tuning to adapt effectively across different tasks or datasets. Another limitation is that our experiments primarily focused on static adversarial behaviors; FedEach has not yet been evaluated under adaptive attack strategies, where malicious clients dynamically change their behavior to evade detection. Addressing such adaptive threats will be an important direction for future research to further strengthen the robustness of the proposed framework. Despite these limitations, the experiment results indicate that FedEach maintains strong performance under adversarial conditions while offering practical efficiency improvements over more complex alternatives.

## 5. Conclusions

This study introduces FedEach, an incentive mechanism for federated learning (FL) that prioritizes fairness, robustness, and efficiency. Unlike conventional approaches, FedEach identifies reliable clients without requiring validation data from the server or a benchmark dataset. Instead, it dynamically selects both evaluators and participants based on client information without identity leakage. Evaluators play a crucial role in assessing the reliability of participants, ensuring that only models from trusted sources contribute to the global model.

A key advantage of FedEach is its ability to fairly evaluate client contributions while maintaining performance, even in adversarial settings where malicious clients form the majority. Experimental results demonstrate that FedEach is particularly effective in such environments and achieves greater efficiency than the Shapley value approach. This efficiency makes it especially well-suited for HAR, where real-time computation and low computational cost are essential. By reducing the overhead associated with model aggregation, FedEach offers a practical solution for deploying FL in resource-constrained HAR applications, such as wearable devices and mobile sensing.

Nevertheless, FedEach has certain limitations. While it effectively evaluates data quality, it does not take data volume into account. This means that a client with a large dataset and another with a much smaller dataset may receive the same rewards if their data quality is equivalent. Additionally, the performance of FedEach depends on setting an appropriate threshold (δ), which requires a proper tuning approach. To address these challenges, future research will focus on incorporating data quantity into the evaluation process and developing a systematic approach for determining the optimal threshold (δ) to further enhance the effectiveness of FedEach. Furthermore, future work will include implementing and evaluating the proposed approach on edge and mobile devices with adaptive attack strategies to demonstrate its practical applicability in real-world, resource-constrained environments.

## Figures and Tables

**Figure 1 sensors-25-03687-f001:**
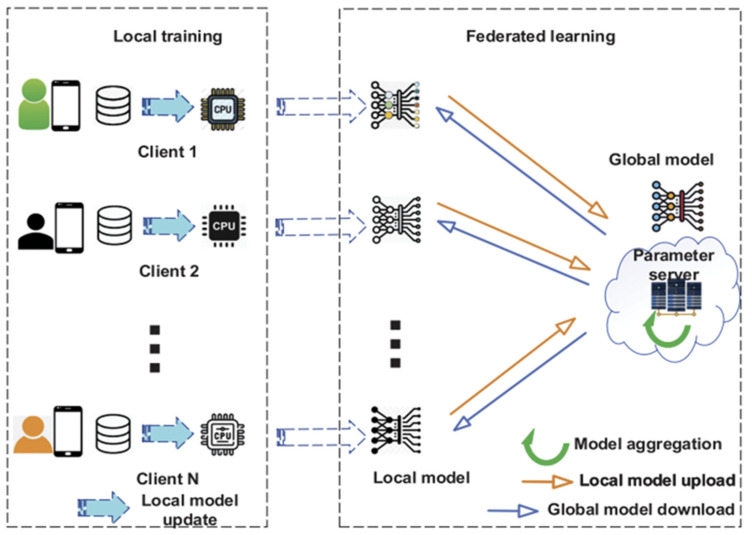
Overview of FL system [9].

**Figure 2 sensors-25-03687-f002:**
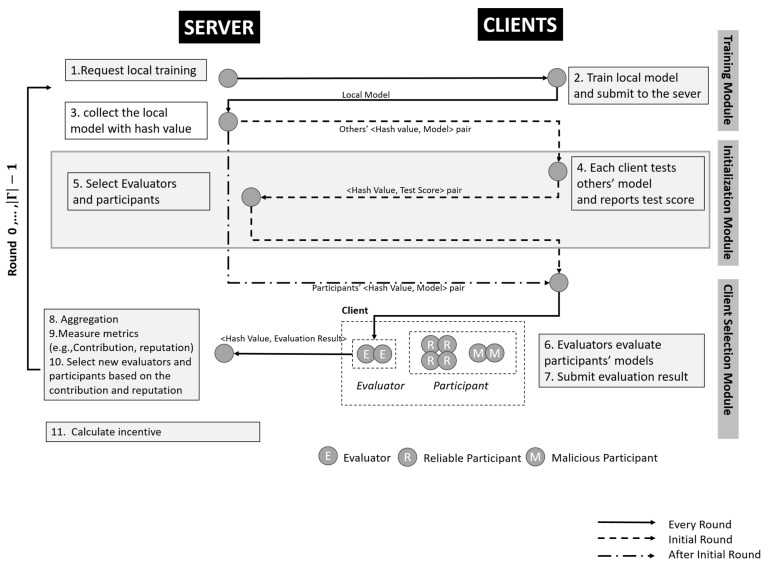
FedEach framework.

**Figure 3 sensors-25-03687-f003:**
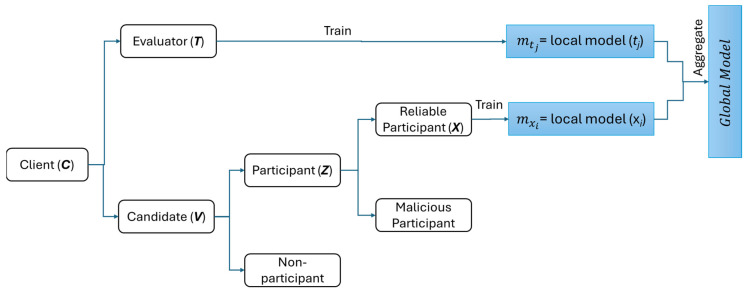
Client categorization mechanism and global model.

**Figure 4 sensors-25-03687-f004:**
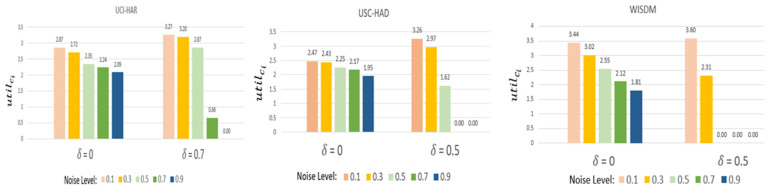
Fairness evaluation based on contribution degree of subject #1.

**Figure 5 sensors-25-03687-f005:**
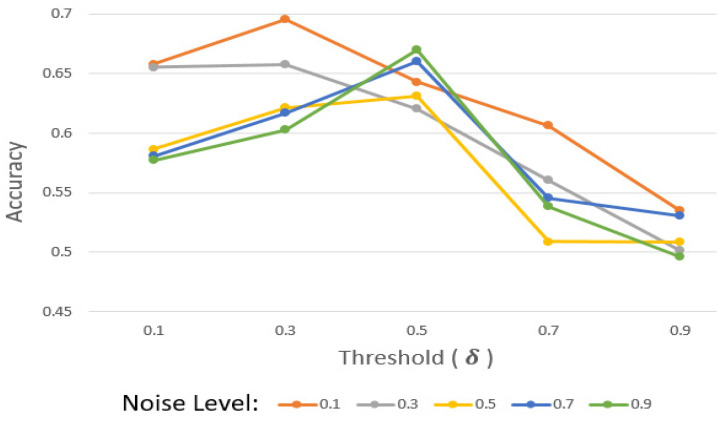
Sensitivity experiment result.

**Table 1 sensors-25-03687-t001:** List of notations.

Notation	Description
${m^{\prime }}_{g}$	The global model
Γ= {0,1,..r,..,$\left|\Gamma \right|-1\}$	The set of rounds
n	The number of clients
$C=\{c_{1},c_{2},..,c_{n}\}$	The set of clients
** *M* ** $=\{m_{c_{1}},m_{c_{2}},..,m_{c_{n}}\}$	The set of local models
k	The number of evaluators
$T=\{t_{1},t_{2},...,t_{k}\}$, T⊂C	The set of evaluators
$V=\left\{v_{1},v_{2},...,v_{n-k}\right\}$, V⊂C	The set of candidates
β	The number of participants
$Z=\left\{z_{1},z_{2},...,z_{\beta }\right\}$, Z⊂V**,**	The set of participants
** *Z* ** ∖ ** *X* **	The set of malicious participants
α	The number of reliable participants
$\;X=\{x_{1},x_{2},...x_{\alpha }\}$ , X⊂ Z	The set of reliable participants
$R_{c_{i}}$	The reputation of $c_{i}$
$\theta_{c_{i},c_{j}}$	The test score of client $c_{i}$ tested by client $c_{j}$
$P_{v_{i}}^{r}$	The importance ratio of $v_{i}$
$E_{z_{i},\;t_{j}}$	The evaluation result of participant $z_{i}$ by evaluator $t_{j}$
${sim}_{c_{i},r}$	The contribution degree of client $c_{i}$ on round r

**Table 2 sensors-25-03687-t002:** The performance (accuracy) of methods.

Dataset	Proportion of Malicious Clients	FedAvg	TIFF	SVS	FedEach
UCI-HAR	<0.5	0.8754	0.8847	0.9234	0.8992
	≥0.5	0.5569	0.6042	0.9152	0.8812
USC-HAD	<0.5	0.5925	0.6562	0.6908	0.6702
	≥0.5	0.3967	0.4939	0.6458	0.6305
WISDM	<0.5	0.6343	0.7284	0.7281	0.7592
	≥0.5	0.4791	0.5368	0.7307	0.7599

**Table 3 sensors-25-03687-t003:** Running time performance comparison.

	TIFF	SVS	FedEach
Running time (s)	102.736	3817.041	308.735

## Data Availability

Data are publicly available.

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
