# Peer review of "FedEach: Federated Learning with Evaluator-Based Incentive Mechanism for Human Activity Recognition"

_sensors, 2025, doi:10.3390/s25123687_

Round 1
Reviewer 1 Report
Comments and Suggestions for Authors
This paper, titled "FedEach: Federated Learning with Evaluator-based Incentive Mechanism for Human Activity Recognition" proposes a federated learning framework with evaluator-based incentive mechanism (FedEach) that offers robustness with no dependency on server-labeled data. Three modules were implemented: training module, initialization module and client selection module.
This paper fits into the scope of the special issue “Wearable Devices for Physical Activity and Healthcare Monitoring”.
The idea of this paper is interesting. However, I have the following concerns.
- Novelty and contribution must be rearranged and edited:
- “(1) Effectively handling malicious clients when they are present” – I did not find any evaluation when malicious clients were effectively handled.
- “(3) Minimizing computational costs” – observed the running time of three client selection methods did not prove effectiveness and computational cost minimization of the proposed FedEach, especially when only UCI-HAR dataset, that was resampled by the authors, and which is more homogenous than the others. Due to mentioned reasons, it seems that the selected resampled UCI-HAR dataset for effectiveness evaluation was prepared accordingly to get better computational cost.
- “We introduce a clients categorization mechanism which differentiates between clients responsible for evaluating others in initial module and client selection module and those recognized as reliable participants” – the authors should read this contribution carefully and try to explain what does it mean? Categorize in initial module and client selection module and were those recognized as reliable participants?
- I think this article lacks innovation, especially asserting “In addition, efficient client selection is important for HAR applications, where real-time processing and low computational cost are essential for practical solution in resource-constrained environments such as wearable devices and mobile sensing platforms” (lines 119-121). While wearable devices mentioned there, the experiment evaluating computational cost was made using an 11th Generation Intel Core i7-1165G7 processor, 401 supplemented by 16GB of RAM and an NVIDIA GeForce MX450 graphics card.
- What performance measure is used in “Table 2. The performance of methods” accuracy or F1-score or other?
- Proportion of Malicious clients in Table 2 is shown as < 0.5 and ≥ 0.5, but in the comments (lines 450-453) “As shown in Table 2, while FedAvg shows lower performance, most methods remain effective when malicious clients are 20% of the total. However, when the proportion of malicious clients increases to 70%, the results change significantly” due to that the Table 2 must be expanded showing results when malicious clients are in ranges, as an example 20, 30, 40, 50, 60, 70, 80.
- What measures does Table 3 provide—ms, sec, min, or hours?
- Why, for the experiment to evaluate efficiency, is the global round 10 and the number of clients 11? The author proposed an advantage of the FedEach method when there are more than 50% malicious clients.
- The proposed approach does not explicitly address adaptive attack strategies where adversaries dynamically change tactics.
- Also, it would add value to the paper if the authors considered a comparison with a baseline approach, regarding the results.
- This paper doesn't contain the discussion section. Ensure that this section is included and properly described to bring out the major findings of this study and compartment with the state-of-the-art (SoTA) results from other researchers.
There are a lot of typographical errors:
- Figure 1 not referenced in the text (line 152).
- Error! Reference source not found (lines 225, 442, 463, 475).
- Figure 1 number is duplicated (line 310).
I suggest the authors carefully read and improve the readability of this article.
Reviewer 2 Report
Comments and Suggestions for Authors
Thank you for submitting your work to Sensors.
Incentive-based federated learning has been widely investigated, and this work has these strengths.
S1. Authors have classified the related works in each important part.
S2. The three main aspects in incentive-based client selection well motivate the overall problem.
S3. This paper uses three different datasets
However, I have a big concern in presentation. I was totally disrupted by unclear explanations, inconsistent notations, and incorrect formatting, and therefore, it was very hard to understand.
W1. Some motivations are vague. For example, in lines 45-46, what does 'getting clients to contribute their data to the FL process' mean?
W2. References are too old. There are many recent incentive-based client selection or federated learning [1]-[3]. This paper should clearly mention what the novelty compared to existing works is.
W3. While this paper solves a client selection problem, there is a trade-off when excluding some clients--excluding malicious clients make FL better, but at the same time, the amount of training data becomes less and confined/biased. Alternatively, we can reduce the weight on malicious clients while increasing the weight of benign users in aggregation.
W4. Please check all notations again. The bolded notations and plain notations are totally different. Please use consistently.
There are very basic comments that should be revised before submitting the paper. Please take this problem sincerely and proofread the entire paper.
C1. There are some typos, missing punctuations, or missing spaces. For example. (GDPR)[1] -> (GDPR) [1] in line 40; [18] [37] -> [18], [37] in line 73; [24][25] -> [24], [25] in line 116; problem ([21], [22]) -> problem [21], [22].; . No bold on [35] in line 411. Also, in line 115, it seems like there are multiple spaces between HAR and have. The equation number becomes a widow word in lines 327-328. I highly recommend authors to proofread.
C2. The resolution of figures is bad. Also, the Algorithm 1 is too hard to read. Too small. Please do not change the width-height ratio of original figures.
C3. There are two Figure 1. Also, some captions are center aligned, while some are left aligned. Also, some are italic, some are bolded, and some are plain font. Again, I highly recommend authors to proofread and check the formats.
C4. What is the blue [17] in line 321?
C5. What are the equations after line 389?
[1] Lin, Yijing, et al. "Incentive and dynamic client selection for federated unlearning." Proceedings of the ACM Web Conference 2024. 2024.
[2] Guo, J., Su, L., Liu, J., Ding, J., Liu, X., Huang, B., & Li, L. (2024). Auction-based client selection for online Federated Learning. Information Fusion, 112, 102549.
[3] Chen, Z., Zhang, H., Li, X., Miao, Y., Zhang, X., Zhang, M., ... & Deng, R. H. (2024). FDFL: Fair and discrepancy-aware incentive mechanism for federated learning. IEEE Transactions on Information Forensics and Security.
Round 2
Reviewer 1 Report
Comments and Suggestions for Authors
The authors answered all questions and improved the article “FedEach: Federated Learning with Evaluator-based Incentive Mechanism for Human Activity Recognition”.
Author Response
We sincerely thank the reviewer for the time to review our manuscript.
Reviewer 2 Report
Comments and Suggestions for Authors
First of all, thank you for addressing all the comments. I had severe concerns in the presentation of this paper, and now I believe it is much better.
Now, I have a feedback regarding the design and the novelty.
- D1) This work utilizes partial clients to evaluate local models, thereby providing robust and computationally efficient method. Please specify this novelty (if so) in the first bullet point of main contributions: 'We propose a novel ...' so why this is novel and beneficial?
- D2) I have a fundamental question--why selecting the evaluator among the clients is beneficial? Why selecting the evaluator matters? Authors have mentioned some reputations and contributions, but it is hard to get the point. Isn't distinguishing the malicious clients more important? I am not sure whether different evaluators are impacting those evaluations.
- D3) Following up the previous comments, Top-k clients based on S_{c_i} are selected as evaluators. I am not sure why selecting evaluators is the higher priority, and the rest of them are training participants. The purpose of evaluator and this work is to filter out malicious clients, so it seems that having the beneficial participants is more important than having the good evaluator. However, the design is focusing on selecting evaluators, which seems counter-intuitive for me.
- D4) Are those evaluation results (E) computed by training data? Also, what kind of evaluation metrics will be better? For generalization, I understand 'Various metrics like F1-score and accuracy' in line 325; however, what is the promising metric for evaluation result?
- D5) Also, it highly depends on the threshold. It seems to select the clients with accuracy (for example) higher than threshold. It depends on the malicious clients' behavior. What happens if malicious clients tend to be benign and sometimes be malicious? Can you provide a concrete threat model of what malicious clients mean? For example, they are always giving a random weight, they have totally different data distribution, or etc.
The biggest concern is about the results. It seems like in most cases, SVS works better than the proposed FedEach. If we can at least generalize that FedEach works better when there are large number of malicious clients. However, FedEach only works better with WISDM dataset. I got the point that the running time is faster than SVS, but I believe the performance of SVS is mostly better. Authors have discussed in the Discussion section, but I am not sure whether the results are enough.
Still, there need some minor edits.
- E1) Please do not forcibly widen or narrow down the figure, e.g., Figures 2, 4, and 5. I may miss some figures, so please check all the figures again.
- E2) Please add punctuations after all equations.
- E3) I mentioned earlier, the bold and plain notations are totally different. For example, in Eq. 1, why U, theta, and j are in bold texts? Also, ( 2 ) and ( 3 ) are in bold texts. Also, in Eq. 4. Furthermore, italic and non-italic are also different, e.g., T and V in algorithm 1 are in italic, while they are non-italic in main texts.
Round 3
Reviewer 2 Report
Comments and Suggestions for Authors
Thank you for addressing all the comments. Lastly, please check the Sensors formatting about figure/table placement and width. Mostly, the figure/table should be placed within the text width area and top aligned.